



**A mass balance-based emission inventory of non-methane volatile**
**organic compounds (NMVOCs) for solvent use in China**
Ziwei Mo [1,2#], Ru Cui [1,2#], Bin Yuan[1,2]*, Huihua Cai[3], Brian C. McDonald[4,], Meng Li[4,5], Junyu
Zheng[1,2] Min Shao[1,2]*
[1] Institute for Environmental and Climate Research, Jinan University, Guangzhou 511443,
China.
[2] Guangdong-Hongkong-Macau Joint Laboratory of Collaborative Innovation for
Environmental Quality, Guangzhou 511443, China.
[3] Guangdong Polytechnic of Environmental Protection Engineering, Foshan 528216, China
[4] Chemical Sciences Laboratory, NOAA Earth System Research Laboratories, Boulder, CO,
USA
[5] Cooperative Institute for Research in Environmental Sciences, University of Colorado,
Boulder, CO, USA
*Correspondance to Prof. Bin Yuan (byuan@jnu.edu.cn) and Prof. Min Shao
(mshao@pku.edu.cn)
[#]These authors contributed equally to this work





## 29 Abstract

Non-methane volatile organic compounds (NMVOCs) are important precursors of ozone
($O_3$) and secondary organic aerosol (SOA), which play key roles in tropospheric chemistry. A
huge amount of NMVOCs emissions from solvent use are complicated by a wide spectrum of
sources and species. This work presents a long-term NMVOCs emission inventory of solvent
use during 2000-2017 in China. Based on a mass (material) balance method, NMVOCs
emissions were estimated for six categories, including coatings, adhesives, inks, pesticides,
cleaners and personal care products. The results show that NMVOC emissions from solvent use
in China increased rapidly from 2000 to 2014 then kept stable after 2014. The total emission
increased from 1.6 Tg (1.2-2.2 Tg at 95 % confidence interval) in 2000 to 10.6 Tg (7.7-14.9 Tg)
in 2017. The substantial growth is driven by the large demand of solvent products in both
industrial and residential activities. However, increasing treatment facilities on the solvent-
related factories in China restrained the continued growth of solvent NMVOCs emissions in
recent years. Rapidly developing and heavily industrialized provinces such as Jiangsu,
Shandong and Guangdong contributed significantly to the solvent use emissions. Oxygenated
VOCs, alkanes and aromatics were main components, accounting for 42%, 28% and 21% of
total NMVOCs emissions in 2017, respectively. Our results and previous inventories are
generally comparable within the estimation uncertainties (-27%-52%). However, there exist
significant differences in the estimates of sub-categories. Personal care products were a
significant and quickly rising source of NMVOCs, which were probably underestimated in
previous inventories. Emissions from solvent use were growing faster compared with
transportation and combustion emissions which were relatively better controlled in China.
Environmentally friendly products can reduce the NMVOCs emissions from solvent use.
Supposing all solvent-based products were substituted by water-based products, it would result
in 37%, 41% and 38% reduction of emissions, OFP and SOAP, respectively. These results
indicate there is still large room for NMVOCs reduction by reducing the utilization of solvent
product and end-of-pipe control across industrial sectors.



## 1 Introduction


Air pollution has caused wide public attention because of its adverse effect on human
health (Nel, 2005). The high concentrations of ozone ($O_3$) and fine particles ($PM_{2.5}$) are the
main reasons for heavy pollution episodes in urban areas (MEEPRC, 2019). As the precursors
of $O_3$ and secondary organic aerosol (SOA), non-methane volatile organic compounds
(NMVOCs) become the key pollutants targeted for priority control (Nishanth et al., 2014;Hao
and Xie, 2018). China is the hotspot of NMVOC emissions across the world. The total NMVOC
emissions have increased rapidly in recent decades (Simayi et al., 2019;Li et al., 2019;Sun et
al., 2018;Wu et al., 2016;Wang et al., 2014;Wei et al., 2011b). Reducing NMVOC emissions is
of utmost importance for tackling air pollution problems in megacities of China (Jin and
Holloway, 2015;Yuan et al., 2013).
There are various anthropogenic sources of NMVOCs emissions including industrial
processes, fossil fuel combustion, biomass burning, traffic emissions, and solvent utilization
(Li et al., 2015). Multiple emission inventories have been established to quantify NMVOC
emissions for China (Li et al., 2019;Sun et al., 2018;Wei et al., 2011b). The total NMVOCs
emissions were estimated to increase from 19.4 Tg in 2005 to 23.2 Tg in 2015 (Wei et al.,
2011b). A more recent inventory suggested that NMVOCs emissions increased from 9.8 Tg to
28.5 Tg between 1990 and 2017 (Li et al., 2019). The unprecedent increase of NMVOC
emissions in China is largely attributed to the fast urban and industrial expansion. In particular,
NMVOC emissions from solvent use sectors are reported to triple over the past three decades,
becoming the largest emission source in China (Li et al., 2019).
Emission estimates for solvent use are challenging because of the wide spectral of
stationary and fugitive sources. Compared with other key NMVOCs sources such as
transportation and fossil fuel combustion, NMVOCs emissions from solvent use have larger
uncertainties among different emission inventories. The estimated emissions were in the range
of 1.9-5.8 Tg from solvent use while were 4.9-6.1 Tg from transportation and 4.8-7.7 Tg from
combustion for the year of 2005 (Li et al., 2019;Sun et al., 2018;Wang et al., 2014;Wei et al.,
2008;Bo et al., 2008;Wei et al., 2011b). The large uncertainty in solvent use emissions are
resulted from different source categories and different emission factors (EFs) in these
estimations. Specifically, coatings are well identified as the emission category in solvent use
source. However, the sub-categories of coatings are inconsistent among different studies (Sun
et al., 2018;Wu et al., 2016;Yin et al., 2015). It is unclear whether the emission inventories
considered all of the industrial sectors associated with coatings. Adhesives are another



important category of solvent use source. Nevertheless, this category was missing in some
emission inventories, or only shoe-making was considered among a number of sub-categories
for adhesives (Sun et al., 2018;Wu et al., 2016;Yin et al., 2015;Bo et al., 2008). In addition,
non-industrial solvent use such as pesticide or domestic solvents were usually not accounted in
the emission inventories (Fu et al., 2013;Bo et al., 2008). Apart from the differences in
categories of solvent use, the emission factors used in different studies varied significantly. For
example, the EFs differed several times for automobile coating (2.43–21.2 kg/vehicle) (Zhong
et al., 2017;Wu et al., 2016;Bo et al., 2008). Emissions from domestic solvent use were always
estimated by emission factor with a unit of kg/capita. However, recent study argued the
accuracy of using national population to estimate the solvent use emissions (Pearson, 2019).
Unlike the EF-based estimation, the mass balance or material balance (MB) approach
provides reliable average emission estimates for specific sources in developing emission
inventory for solvent use (US EPA, 1995). This technique involves quantification of chemical
material flows going into and out of a process, where the total discharges to the environment
are estimated by input and output information based on the mass conservation principle. The
MB technique was used to update NMVOCs emission estimates for solvent products in the
United States, which were validated by ambient NMVOCs measurements (McDonald et al.,
2018). The successful application of the MB technique for the solvent-related sources provides
important support in developing more accurate emission inventories. Currently, there is still
lack of NMVOCs emission inventories specialized in solvent use in China. In view of large
discrepancies among different studies, re-evaluation of NMVOCs emission estimates are
needed for solvent use in China using the MB technique.
This study focuses on six categories of solvent products used in residential and industrial
activities including coatings, inks, adhesives, pesticides, cleaners and personal care products.
The MB technique is adopted to estimate NMVOCs emissions from these solvent products
between 2000 and 2017 in China. Incorporating the source profiles, speciated NMVOCs
emissions for each solvent product are obtained. Estimated NMVOCs emissions from solvent
use in this study are compared with other studies and other sources. Finally, implications for
NMVOCs emission abatement in China are discussed in terms of ozone formation potential
(OFP) and secondary organic formation potential (SOAP).



## 2 Methods and data

### 2.1 Emission estimation



Six types of organic solvent products are considered in this study, including coatings,
inks, adhesives, pesticides, cleaners, and personal care products. Coatings, adhesives and inks
are further classified based on application fields and/or technologies (level 2), solvent types
(level 3). Personal care products are divided into four sub-categories: hair and body cares,
perfumes, skin cares, and other cosmetics. Pesticides include herbicides, insecticides,
bactericides and other pesticides. Cleaners include laundry, dishwashing, surface cleaners, and
industrial detergents.
Organic compounds in solvent products have different volatilities, which can be
characterized by effective saturation concentration $C_*$. Organic compounds can be classified
into three categories according to the range of effective saturation concentration, namely high-
volatility organic compounds (VOCs: $C_* > 3 \times 10^6$ μg m$^{-3}$), intermediate-volatility organic
compounds (IVOCs: $C_* = 0.3$ to $3 \times 10^6$ μg m$^{-3}$) and semi-volatile organic compounds
(SVOCs: $C_* < 0.3$ μg m$^{-3}$). Hence, organic solvent content in products is divided into VOCs
and S/IVOCs, considering volatilization of VOCs and S/IVOCs respectively. The mass
balance approach, also called material balance is adopted to estimate NMVOCs emitted by
organic solvent products, as detailed in McDonald et al. (2018). The total NMVOCs
emissions from solvent products are estimated by Equation (1):
$$E_n = \sum_i A_{i,n} \cdot (W_{VOC,i} \cdot VF_{VOC,i} + W_{S/IVOC,i} \cdot VF_{S/IVOC,i}) \cdot (1 - C_n \cdot \eta_{avg}) \qquad (1)$$
where $E_n$ (g) is the total NMVOCs emissions from all solvent products in a certain year $n$;
$A_i$ (g) is the consumption of product $i$; $W_{VOC,i}$ (g solvent g$^{-1}$ product) is the average VOC
content while $W_{S/IVOC,i}$ is the average S/IVOC content in product $i$; $VF_{VOC,i}$ (g emitted g$^{-1}$
dispensed VOC) and $VF_{S/IVOC,i}$ (g emitted g$^{-1}$ dispensed I/SVOC) are volatilization fractions
of VOCs and S/IVOCs for product $i$. $C_n$ is the percentage of treatment facilities installed in
the industrial sector in the year $n$; and $\eta_{avg}$ is the average reduction coefficient induced by
treatment facilities. Noted that only the control of NMVOCs emissions from industrial solvent
use is considered in this study.
Product consumption data ($A_i$) are mainly collected from official statistical yearbook.
Consumption of adhesive is from China Chemical Industry Yearbook (CPCIA, 2000-2016).
However, formaldehyde-type adhesives is not reported in the yearbook in most cases.
Considering that formaldehyde-type adhesive is mainly used in artificial board manufacturing,



we assumed a linear relationship between formaldehyde-type adhesive consumption and the
artificial board yield, and estimated the missing data of formaldehyde-type adhesives based on
this linear relationship (seeing Figure S1). Consumption of ink, cleaner and personal care are
from China Light Industry Yearbook (CNLIC, 2001-2018). It should be noted that
consumption data for personal care products are not directly available in the yearbook, which
are estimated from dividing sales of the product by unit price. Consumption of coating are
from China Paint and Coating Industry Annual (CCIA, 2000-2017). There are four data
sources collected for pesticide (Figure S2), we choose China Crop Protection Industry
Yearbook (CCPIA, 2001-2017) and Duan (2018).

161        VOCs contents ($W_{VOC}$) in products are derived from various domestic and international

regulations or standards. Taking architectural coatings for example, VOCs contents of
architectural coatings are based on GB18582-2008 and GB24408-2009. More details about
VOCs contents in products are shown in Table S1-S5. S/IVOCs contents ($W_{S/IVOC}$) are
derived from ratios of VOCs and S/IVOCs to organic solvent. The equation of S/IVOCs
contents is as follows:
$$W_{S/IVOC,i} = \frac{f_{S/IVOC,i}}{f_{VOC,i}} \cdot W_{VOC,i} \qquad (2)$$
where $f_{VOC,i}$ (g VOC g$^{-1}$ solvent) and $f_{S/IVOC,i}$ (g S/IVOC g$^{-1}$ solvent) are fractions of
organic solvents as VOCs and S/IVOCs in product $i$. The parameters of $f_{VOC,i}$, $f_{S/IVOC,i}$,
$VF_{VOC,i}$ and $VF_{S/IVOC,i}$ in Eqation (1) and (2) are referred to McDonald et al. (2018).

171        Monte Carlo analysis is applied to estimate uncertainty of annual emissions. The

variation coefficients of activity data are determined by the empirical values depending on the
source of activity (Wei et al., 2011a). Specifically, uncertainty is set to be ± 30% if data are
directly from official statistics; uncertainty is assumed to be ± 80% if activity data is
estimated from other statistical information or reports. Uncertainty of $W_{VOC}$ is based on
VOC content raw data (Table S1-S5). Uncertainty of $W_{S/IVOC}$ is referred to that of $W_{VOC}$.
Specific classification of solvent use and various parameters are shown in Table S6.
**2.2 Spatial allocation**

179        Total NMVOCs emissions of solvent use in China are allocated to provincial level based

on a top-down approach. The proxy variables of cultivated land area, disposable income, sales
value and building area completed in different provinces are used for allocation (Table S7).
Then, the provincial emissions are calculated using Equation (3).





$$E_m = \sum_i \frac{T_m}{\sum_m T_m} \cdot E_i \tag{3}$$

where $E_m$ is the emissions from solvent use in province $m$; $E_i$ is the emissions of solvent
product $i$ at the nation level; and $T_m$ is the cultivated land area, disposable income, sales
value or building area completed in province $m$.

**2.3 Estimation of speciated emissions, OFP and SOAP**

Speciated NMVOCs emissions are calculated by allocating the source profiles to the
corresponding emission sources. Source profiles of solvents use used in this study are obtained
by combining domestic profiles and foreign profiles (Li et al., 2014). Detailed methods of
compiling the composite profiles of architectural coating, furniture coating, automobile coating,
other coating, offset printing ink, letterpress printing ink, gravure printing ink, other printing
ink, shoemaking adhesive, and herbicide are provided in Text S1 and Figure S3-12. For products
lacking domestic source profile, foreign source profiles were directly used.
The emissions of individual NMVOCs species can be estimated by multiplying the total
NMVOCs emissions by the weight percentage of each species, as shown in Equation (4).

$$E_j = \sum_i E_i \times f_{i,j} \tag{4}$$

where $E_j$ is the emissions of species $j$ from all sources; $E_i$ is total NMVOCs emissions from
organic solvent product $i$; $f_{i,j}$ is the weight percentage of species $j$ in the emission of
product $i$.
The OFP represents the maximum ozone contribution of NMVOCs species, which can
help identify the key reactive species and sources for ozone formation. The OFP of individual
species can be calculated by Equation (5).

$$OFP_j = E_j \times MIR_j \tag{5}$$

where $OFP_j$ is the OFP of species $j$; $E_j$ is the emissions of species $j$ and $MIR_j$ is the
maximum incremental reactivity (MIR) of species $j$ (Carter, 2010).
The SOAP indicates the SOA formation ability of different NMVOCs species, which
can be characterized by SOA yield (McDonald et al., 2018). Then, the SOAP of individual
species can be calculated using Equation (6).

$$SOAP_j = E_j \cdot Y_{SOA,j} \tag{6}$$

where $SOAP_j$ is the SOAP of species $j$; $Y_{SOA,j}$ is the SOA yield of species $j$.



## 3 Results

### 3.1 Control of NMVOCs emissions

The control on NMVOCs emissions from solvent use were not widely implemented in China before 2010. To slow down the rapid growth in NMVOC emissions in China, *the Action Plan for Air Pollution Prevention and Control* issued by the State Council of China in 2013 are explicitly proposed to implement control of NMVOCs emissions from the most important NMVOCs industrial sources, including organic chemistry industries, surface coating industries, printing industries and so on. As the result, control measures are required to be installed in NMVOCs emitting industrial facilities related to solvent use in China. The percentage of solvent use industrial facilities with treatment devices ($C_n$ in Equation 1) increases quickly in the recent years. Based on detailed filed survey in the centers of solvent product manufacturing in China - Yangtze River Delta (YRD) (Lu et al., 2018;Yang et al., 2017) and Pearl River Delta (PRD) region (Gao et al., 2015;Cai, 2016), solvent use factories with treatment facilities reached almost 50% in 2015. Considering that exhaust gas treatment level of different regions are close (MEEPRC, 2017), this value is adopted to represent the whole country. Drastic increase (by a factor of over 15) of annual production value for organic exhaust gas treatment industry were also recorded between 2013 and 2017 (EGPCCEPIA, 2008-2017). Referring 50% of solvent use factories installing treatment facilities in 2015 and the fast growth of production value for organic exhaust gas treatment devices, we estimated the percentage of treatment facilities installed in the industrial solvent sector for other years, assuming slow (1%), moderate (3.3%) and fast (10-15%) increase rate of the percentage before 2010, between 2010-2013 and after 2013, respectively (Figure 1). We then used the estimated percentage with treatment facilities as $C_n$ in Equation (1).

For the treatment facilities, the control efficiency varied significantly by adopted different technology, such as adsorption, absorption, catalytic combustion, photolysis and plasma. Here, we determined averaged control efficiency ($\eta_{avg}$) based on the market shares of VOC control techniques ($f_n$) and their control efficiency ($\eta_n$) (Table S8) by Equation (7).

$$\eta_{avg} = \sum_n f_n \times \eta_n \qquad (7)$$

The market share of NMVOC control techniques and their control efficiency were collected from field surveys in the YRD and PRD regions (Lu et al., 2018;Cai, 2016). The average control efficiency was determined to be about 43% based on the two surveys. Finally, the overall effective control efficiency ($C_n \times \eta_{avg}$) for different years is shown in Figure 1. The

overall efficiency for industrial solvent use facilities increased moderately before 2010, with
values of less than 5%. It increased faster from 2013 at 9% and reached 30% in 2017.
**3.2 Total NMVOCs Emissions**
The estimated annual emissions of solvent NMVOCs in China between 2000 and 2017
are shown in Figure 2. NMVOCs emissions were found to continuously increase from 2000 to
2014 but reached a plateau afterwards. The total NMVOCs emissions were estimated to be 1.6
Tg (1.2-2.2 Tg at 95 % confidence interval) in 2000, increasing (by a factor of 6.7) to 10.6 Tg
(7.7-14.9 Tg) in 2017. We also considered another two scenarios to investigate the effect of
control measures in reduction of NMVOCs emissions: emission without any control (scenario
1); and emission if control efficiency is compromised by 50% (scenario 2), which represents
widespread lack of maintenance in NMVOCs treatment facilities and/or stopping running of
treatment facilities to save cost. In both scenarios, continuous growth of NMVOCs emissions
from 2000 to 2017 was observed. NMVOCs emissions in 2017 for the two scenarios were
estimated to be 13.1 Tg and 11.8 Tg, significantly higher than the estimates considering the real
maintenance practice of NMVOC control (i.e. the best estimate). These results indicate the
importance of NMVOC control measure in preventing the fast increase of NMVOCs emissions
from industrial solvent use. The overall effective control efficiency in industrial NMVOC
emissions was estimated to be only 30%, leaving significant room to further increase the overall
control efficiency. This would be more easily achieved by adopting the NMVOCs control
techniques with better control efficiency (e.g. catalytic combustion), as most of the industrial
NMVOC facilities are already with treatment facilities (70% in 2017).
On the basis of the best estimate of NMVOCs emissions, coating was the major
contributor to the total solvent NMVOCs emissions in most years (42%-58% of total emission
during 2000-2017). The NMVOCs emissions from coatings reached 6.1 Tg in 2017, an increase
of 5.3 Tg (660%) compared with those (0.8 Tg) in 2000. Personal care products (emitting 2.2Tg
NMVOCs in 2017) ranked the second in the contributions to NMVOCs emissions, which,
however, were usually lack of comprehensive estimates in previous inventories (Wu et al.,
2016;Fu et al., 2013;Wei et al., 2008;Bo et al., 2008). Following were adhesives emissions,
increasing from 0.3 Tg in 2000 to 1.6 Tg in 2017. It was commonly used in the shoemaking and
furniture manufacturing which were fast-developing industries in China. Pesticides were also
an important source of NMVOCs emissions from solvent use, accounting for 3%~10% of total
emissions. Apart from coatings, personal care products, adhesives and pesticides, NMVOCs
emissions from inks and cleaning agents accounted for a small proportion (2%~5%) of total



solvent NMVOC emissions. In particular, productions of cleaners were large in China,
approaching 13 Tg in 2017. However, in view of low solvent contents of most cleaning agents
and their treatment processes (e.g. most of S/IVOCs entered sewage), NMVOC emissions only
took up less than 1% of the cleaning agent productions (0.005 g/g in 2017). Emissions from
industrial solvent use were dominant (56%) in 2017 due to the huge industrial demand for
adhesives and coatings in China. About 82% of NMVOCs from non-industrial were caused by
architectural coatings and personal care products. In summary, coatings, personal care products,
adhesives and pesticides were four major NMVOCs emission products, accounting for more
than 95% of total emissions, suggesting that these products are key solvent sources for
NMVOCs control in China.

### 287    3.3 Provincial emissions

Provincial emissions and their contributions by source in 2017 are shown in Figure 3.
Jiangsu, Shandong and Guangdong provinces contributed the most in China, emitting 1.3 Tg
(12.2% of solvent NMVOC emissions in China), 1.1 Tg (10.1%) and 1.0Tg (9.7%) NMVOCs,
respectively. Coatings dominated in the emissions of the three provinces, accounting for 65%,
60% and 61% of solvent NMVOC emissions in Jiangsu, Shandong and Guangdong. Similarly,
with coatings as the major contributor, Zhejiang, Henan, Hubei, Sichuan, Fujian, Hunan, Anhui
were also on the top ten list of NMVOC emissions. These provinces are mainly located in the
eastern and middle areas of China, where the economics are developing fast and industrial
activities are densely distributed, which are driving factors for tremendous NMVOCs emissions.
By contrast, Xinjiang, Gansu, Ningxia, Qinghai and Xizang, located in the vast western inland
areas with a sparse population and slower economic growth, generated no more than 0.1 Tg in
2017. In these slower developing provinces, personal care products and pesticides emissions
comprised a relatively large part because of lower contribution from industrial sectors. These
features suggested that the NMVOCs emissions in different provinces of China were
significantly associated with their developments of urbanization and industrialization.

### 303    3.4 Speciated NMVOCs emissions, OFP and SOAP

The NMVOCs functional group pattern and the top 10 species in VOCs emissions in
2017 are illustrated in Figure 4. Of the total emissions (10.6 Tg), OVOCs and alkanes were the
main components, accounting for 42% and 28%, respectively (Figure 4a). They were followed
by aromatics (21%), halocarbons (3%), and alkenes (2%). The top three NMVOCs groups were
similar to those in a previous emission inventory, with OVOCs (more than 35% of total



emissions), aromatics (24%) and alkanes (21%) as the main NMVOC groups (Wei et al., 2008).
The large amount of alkanes mainly came from coatings and adhesives (Figure S13),
contributing 1.3 Tg and 1.0 Tg of total alkanes, respectively, in 2017. OVOCs were dominated
by coatings (2.4 Tg) and personal care products (1.4 Tg). Of total aromatics emissions (4.4 Tg),
near 88% of the emissions were attributed to coatings. For the individual species (Figure 4b),
the top 10 species of emission were ethanol (1.1 Tg), ethyl acetate (0.8 Tg), toluene (0.5 Tg),
acetone (0.4 Tg), m/p-xylene (0.4 Tg), styrene (0.3Tg), isobutane (0.3 Tg), propane (0.3 Tg),
ethylbenzene (0.3 Tg) and o-xylene (0.2 Tg). As a common component of daily-used solvent
products, ethanol was the largest emission species from personal care products and cleaner. This
suggests that solvent use might be another important emission source of ethanol in urban areas
in addition to vehicle emissions for the regions using ethanol-containing gasoline (Khare and
Gentner, 2018;de Gouw et al., 2012).
Comparison of emissions, OFP and SOAP in 2000 and 2017 are shown in Figure 5 in
terms of NMVOCs groups and solvent use categories. NMVOCs emissions from solvent use
increased from 1.6 Tg in 2000 to 10.6 Tg in 2017 by a factor of 6.7. OFP and SOA increased
from 3.2 Tg to 21.3 Tg (by a factor of 6.6) and from 0.06Tg to 0.39 Tg (by a factor of 6.7),
respectively. The similar growth factors among emissions, OFP and SOAP indicate relatively
small effects of emission structure and reactivity of NMVOCs. The largest group of OFP was
aromatics, accounting for 54% of total OFP in 2017 (Figure 5a). OFP from OVOCs and alkanes
took up only 27% and 14% respectively, though their emissions are higher. OFP of alkenes only
contributed 4%. As for SOAP, aromatics were also the main contributor (38%). It was followed
by alkanes (31%), OVOCs (12%) and alkenes (6%). The differences in emissions, OFP and
SOAP contributions from NMVOCs groups are due to differences in MIR and SOA yields of
NMVOCs species. Figure 5b shows OFP and SOAP from solvent use categories. Coatings are
the major contributors to OFP and SOAP, accounting for 68% and 58% respectively in 2017.
The contributions of adhesives and personal care products to OFP (14%) and SOAP (16% and
15%) are similar. OFP and SOAP from ink, pesticide and cleaner are less than other three
categories, with the total not exceeding 10%.

## 4 Discussions

### 4.1 Comparison with other studies

NMVOCs emissions from solvent use in this study are compared with EIs in literature
(Figure 6), including Regional Emission inventory in Asia (REASv3.1) (Kurokawa et al., 2013) ,



Emission Database for Global Atmospheric Research (EDGARv4.3.2), MEIC (Li et al., 2019),
Sun EI (Sun et al., 2018) and Wu EI (Wu and Xie, 2017;Wu et al., 2016). Our estimates were
peaked in 2014, the same with REASv3.1 whose emissions, however, were much higher.
Emissions in EDGARv4.3.2 were significantly higher than our work in early 2000s. However,
with much higher annual growth rate of 12% in our work, emissions surpassed those in
EDGARv4.3.2 after 2011. The differences between our work and two foreign studies are mainly
due to different emission factors and source classification. Compared with the domestic long-
term EIs in China, our results were much higher than Sun EI (from 1.6 Tg in 2000 to 5.0 Tg in
2015; 8%) but very close to MEIC (from 2.3 Tg in 2000 to 11.9 Tg in 2017; 10%). However,
MEIC showed continuously increasing trend after 2014 but a plateau of NMVOCs emissions
was found in this study. It is probably because MEIC did not consider the control of NMVOCs
in recent years.

353       For the single year estimates, our results were higher than those in Bo et al. (2008) and

Wu et al. (2016), and lower than Wei et al. (2008). The reasons for differences in previous
studies are because different source categories were included and different EFs/activity data.
Bo et al. (2008) and Wu et al. (2016) did not include the emissions from adhesives and method
used to estimate personal care emissions were different between our work and two previous
works. EFs of solvent based adhesives and inks in Wei et al. (2008) were higher than estimation
parameters in our work. Pharmaceutical production and edible oil production were included in
Wei et al. (2008) but not in our work. Different types of activity levels and emission factors also
resulted in the discrepancy in EIs.

362       In order to further examine the emission differences, we compared the emission estimates

between this study and other two EIs, MEIC and Sun EI, with available sub-categories of
solvent use (Figure 7). Coatings emissions in this study agreed well with MEIC but much higher
than Sun EI (Figure 7a). It was attributed that coating emissions in Sun EI only considered
architecture, vehicle and home appliances coating, but ignored other coating industries (can
coating, magnet wire coating, ship painting). Ink emissions were much larger in MEIC, while
similar results were found for Sun EI and this study (Figure 7b). For adhesives, the estimated
emissions in this study were higher than MEIC after 2006 (Figure 7c). This might be attributed
to different emission factors and increased consumption of formaldehyde-type adhesives, which
is missing from the statistical yearbook. Note that adhesives were not included in Sun EI.
Pesticides emissions showed a similar trend between Sun EI and this study, but lower than
estimates in MEIC (Figure 7d) and there is a significant decrease in 2017 in our work due to
that the production of pesticides has decreased and export has increased (Figure S2). For





personal care products, this work estimated much larger emissions than MEIC and Sun EI
(Figure 7e). MEIC and Sun EI estimated domestic solvents emissions using emission factors
with a unit of kg per capita and population data. Therefore, the emission trends of personal care
products in MEIC and Sun EI followed the increasing pattern of China's population (Figure 7e).
In contrast, this study adopted consumption data of personal care and solvent contents used in
chemical products for estimation. Disposable income of households kept similar growth with
our results of the emissions from personal care, suggesting more reasonable estimates in this
study.

### 383     4.2 Comparison with other sources

Figure 8 compares NMVOC emissions from solvent use with other sources (including
transportation, industrial process, and combustion) in MEIC (Li et al., 2019). Solvent use was
not the most significant emission source in the early 2000s, which was lower than combustion
and transportation emissions (Figure 8a). However, solvent use emissions overtook after 2011,
becoming the largest emission sources compared with other sources. It kept growing fast during
2005-2013 and reached a plateau after 2014. It was mainly attributed to the significant industrial
expansion in China over the past decades. This also resulted in continuous increase of NMVOC
emissions from industrial process revealed by MEIC. In contrast, combustion in MEIC and
transportation in MEIC exhibited a decline in past decade, mainly because of the stringent
control of NMVOC emissions from fuel combustion in industrial and on-road vehicles. We also
looked details into the increasing rate of different sources (Figure 8b). Compared with 2000,
solvent use emissions increased by 570% in 2017 in this work, 270% for industrial process in
MEIC in 2017. The transportation and combustion emissions increased (within 50% compared
with 2000) less and then decrease to emission level of 2000 reported by MEIC (Li et al., 2019).
The rapid increase of solvent use emissions over 2000-2017 suggested that solvent use
emissions had become one of the most prominent sources of NMVOCs emissions. It has the
most significant emission reduction potential rather than other sources such as transportation
and combustion in China.

### 402     4.3 Implications for NMVOCs control

In order to reduce NMVOCs emission from solvent use, water-based products, which are
regarded as environmentally friendly, can substitute solvent-based products in China. Taking
the 2017 data as an example, we assumed that all solvent-based products were replaced by
water-based products and evaluated NMVOCs emission reduction effect. Figure 9 shows the



reduction of emissions, OFP, and SOAP after replacing solvent-based by water-based products.
NMVOCs emissions are reduced by 37% from 10.6 to 6.7 Tg, while OFP and SOAP are reduced
by 41% from 21.3 to 12.6 Tg and 38% from 0.39 to 0.24 Tg, respectively. Coatings contribute
most to NMVOCs emission, OFP and SOAP reduction because solvent-based coatings are
dominant in industrial coatings at present. The reductions of adhesives and inks emissions, OFP
and SOAP are minor due to the wide use of water-based solvent in these products. In terms of
species groups, the top three groups of NMVOCs emission reduction are OVOCs (reducing 1.5
Tg, 14% of total emissions), aromatics (1.2 Tg, 11%) and alkanes (1.0 Tg, 9%). However, the
top three groups of OFP and SOAP reduction are different from those of emissions. Aromatics
(reducing 5.8 Tg, 27% of total OFP), OVOC (1.8 Tg, 8%) and alkanes (1.0Tg, 5%) are main
groups of OFP reduction, while aromatics (reducing 0.08 Tg, 20%), alkanes (0.04 Tg, 10%) and
OVOCs (0.01 Tg, 3%) contribute most to SOAP reduction. In general, replacing solvent-based
by water-based products would benefit the NMVOCs reductions with coatings and aromatics
abatement being effective in OFP and SOAP reduction.

## 421     5 Conclusions

NMVOCs emission inventory including six categories solvent products are developed
for the period of 2000–2017, based on the mass balance method. Solvent use NMVOCs
emissions were estimated to increase from 1.6 Tg (1.2-2.2 Tg at 95 % confidence interval) in
2000 to 10.6 Tg (7.7-14.9 Tg) in 2017. However, emissions leveled off between 2014 and 2017.
The control efficiency of industrial solvent NMVOCs was only 30% in 2017, and there is still
room for improvement in NMVOCs control efficiency. Future emissions of NMVOCs from
solvent use depend on product consumption, product solvent type and overall control efficiency.
The major sources of NMVOCs emissions in solvent products were coatings, adhesives and
personal care products, together contributing more than 90% of the total emissions. Industrial
solvent emissions were dominant due to widely use of adhesives and coatings across the
industrial sectors. Personal care products and architectural coatings were major sources of non-
industrial solvent emissions. The regional distribution of VOCs emissions was highly
associated with the level of economic development. Economically developed provinces in
China contributed much more solvent NMVOCs than underdeveloped areas. Alkanes and
OVOCs were the main species emitted from solvent use, followed by aromatics. They were
mainly emitted from adhesives, coatings and personal care products. The top 10 emission
species were ethanol, ethyl acetate, toluene, acetone, m/p-xylene, styrene, isobutane, propane,



ethylbenzene and o-xylene.
OFP and SOAP from solvent use were 21.3 and 0.39 Tg in 2017 respectively. Alkanes,
alkenes, and aromatics were major contributors to OFP and SOAP. Compared with other solvent
use categories, reducing coating emissions is more effective in controlling $O_3$ and SOA
pollution. Emissions from solvent use are growing fastest as transportation and combustion
emissions are well controlled. Low solvent products can reduce NMVOCs from solvent use in
China. Assuming all solvent-based products are replaced by water-based products in 2017,
emissions, OFP and SOAP were reduced by 3.9 Tg, 8.7 Tg and 0.15 Tg respectively, accounting
for more than 35%. It is suggested that there is still room for NMVOCs emission reduction
from solvent use in China.

**Acknowledgements**
This work was supported by the National Key R&D Plan of China (grant No. 2019YFE0106300,
2018YFC0213904, 2016YFC0202206), the National Natural Science Foundation of China
(grant No. 41877302), Guangdong Natural Science Funds for Distinguished Young Scholar
(grant No. 2018B030306037), Key-Area Research and Development Program of Guangdong
Province (grant No. 2019B110206001), Guangdong Soft Science Research Program
(2019B101001005), and Guangdong Innovative and Entrepreneurial Research Team Program
(grant No. 2016ZT06N263). This work was also supported by Special Fund Project for Science
and Technology Innovation Strategy of Guangdong Province (Grant No.2019B121205004).

**Data availability**
Data is available from the authors upon request

**Competing interests**
The authors declare that they have no conflicts of interest

**Author contributions**
BY and MS designed the research. ZM, RC, BY, HC, BM contributed to data collection. ZM
and RC performed the data analysis, with contributions from BY, HC, BM, ML, JZ and MS
ZM, RC and BY prepared the manuscript with contributions from other authors. All the athors
reviewed the manuscript.



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



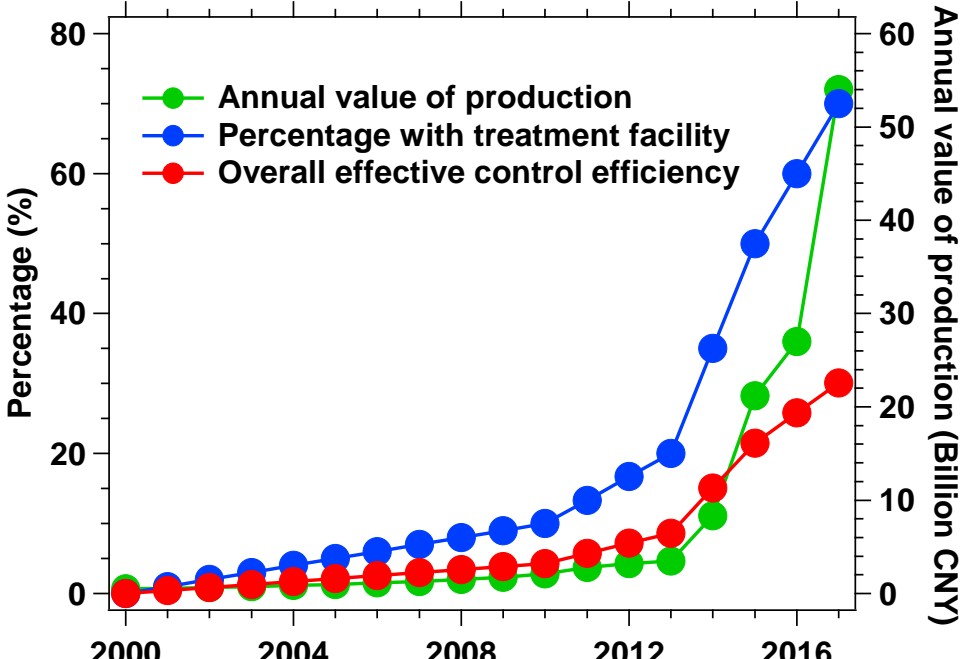

**Figure 1.** The annual value of production for organic exhuast gas treatment industry, percentage with treatment facility installed for solvent-relating factories and the overall effective control efficiency for NMVOCs emissions from industrial solvent use factories in China.



**Figure 2.** (a) Annual NMVOCs emissions from solvent use from 2000 to 2017 in China. (b)
Three scenarios are considered: emission without control; emission if control is compromised
considering the lack of manual maintenance of facility; emission considering the real
maintenance practice of NMVOCs control.

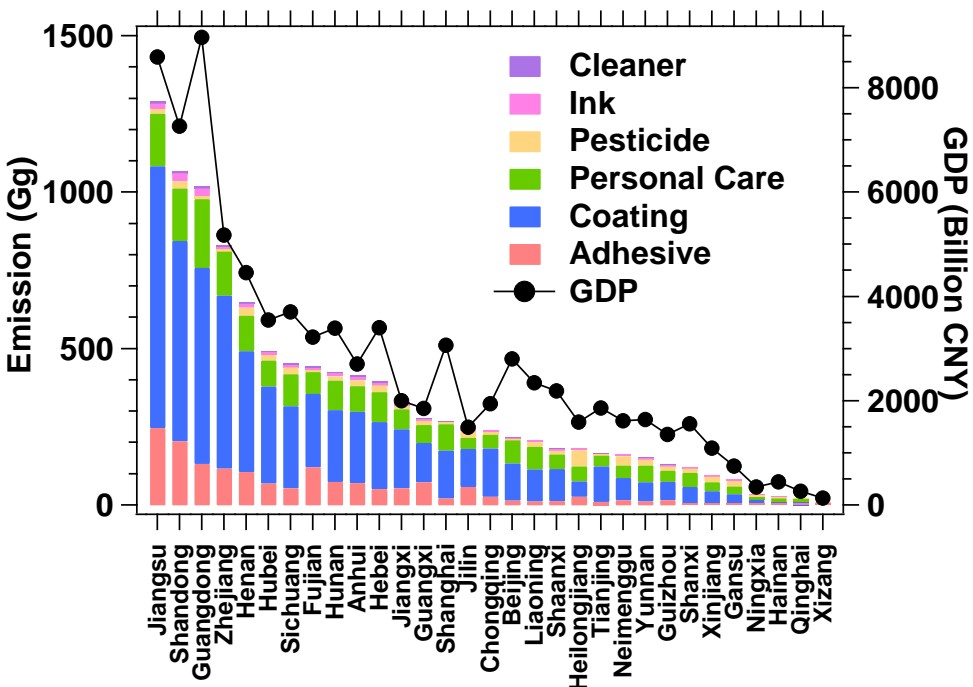

**Figure 3.** Solvent use NMVOCs emissions from different provinces of China in 2017

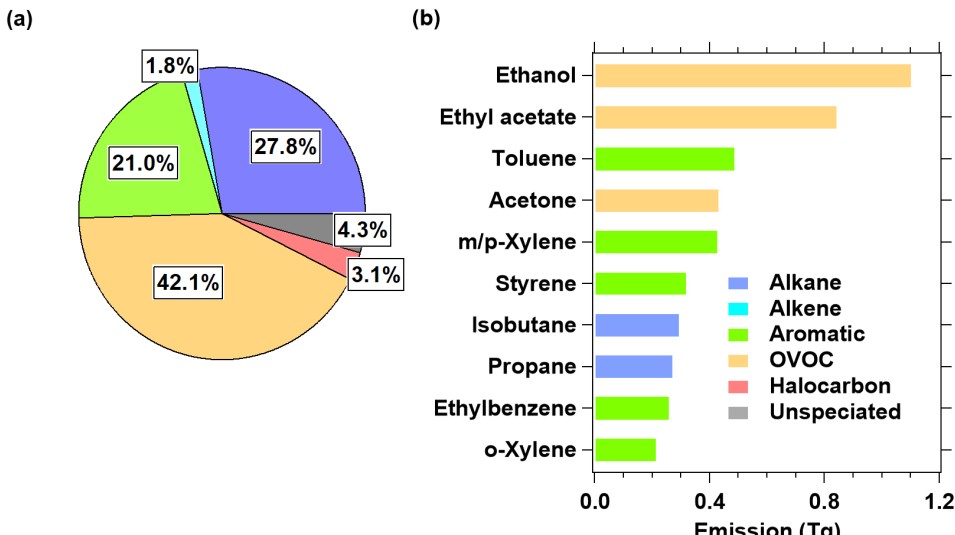

**Figure 4.** (a) Contributions of different NMVOCs groups to total NMVOC emissions, and (b) the top 10 species in NMVOCs emissions in 2017 from solvent use.





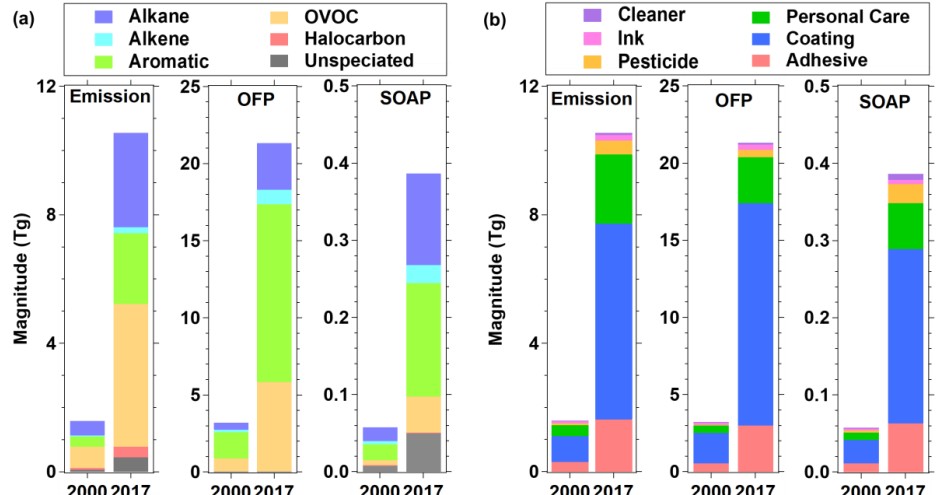

**Figure 5.** Contributions from (a) different source categories and (b) different NMVOCs groups to emissions, OFP, and SOAP of NMVOCs from solvent use in 2000 and 2017.

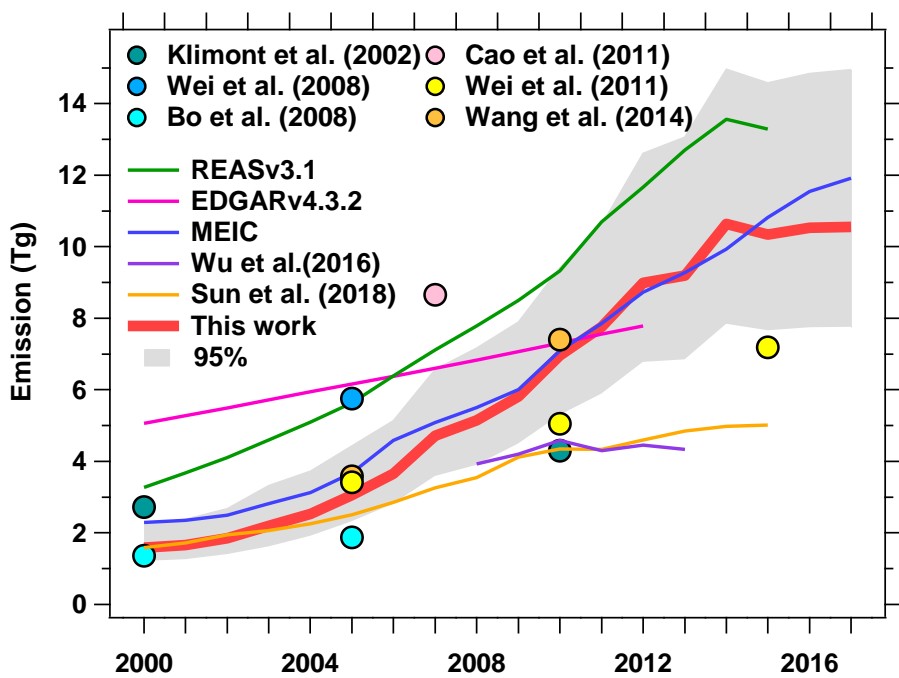

**Figure 6.** Comparison of NMVOCs emissions from solvent use between this study and previous estimates**.**

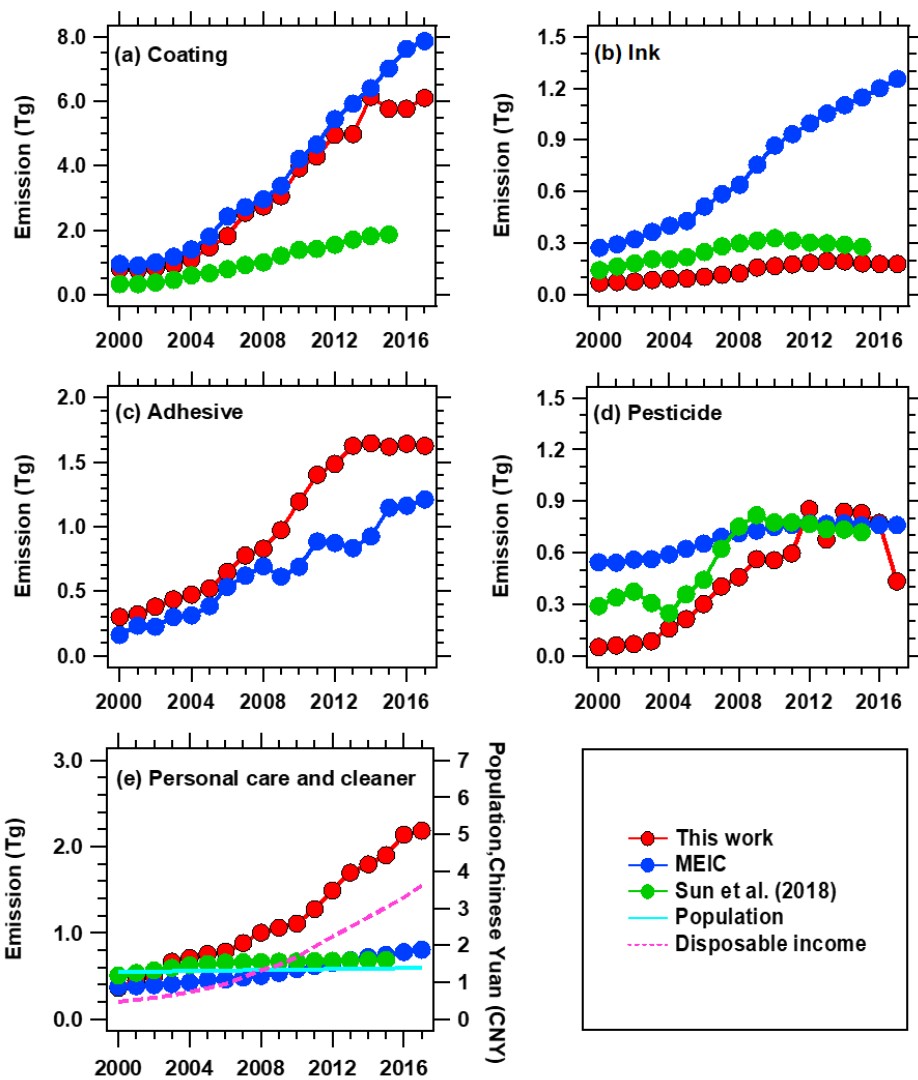

**Figure 7.** Comparisons of emission estimates for (a) coatings, (b) inks , (c) pesticides, (d) adhesives, (e) personal care products, and cleaners (industrial detergents are not included in this figure) between this work and other studies (Li et al., 2019;Sun et al., 2018). Also shown are population (billion) and disposable income of households ($10^{13}$ CNY).



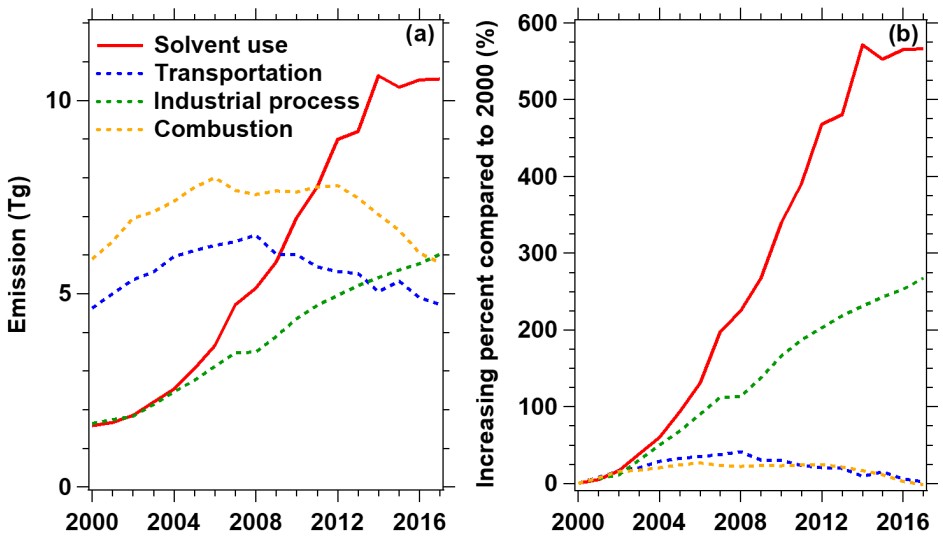

**Figure 8.** Comparisons of (a) NMVOCs emissions and (b) their increasing percentage compared to 2000 from solvent use (this study) and other sources (MEIC).

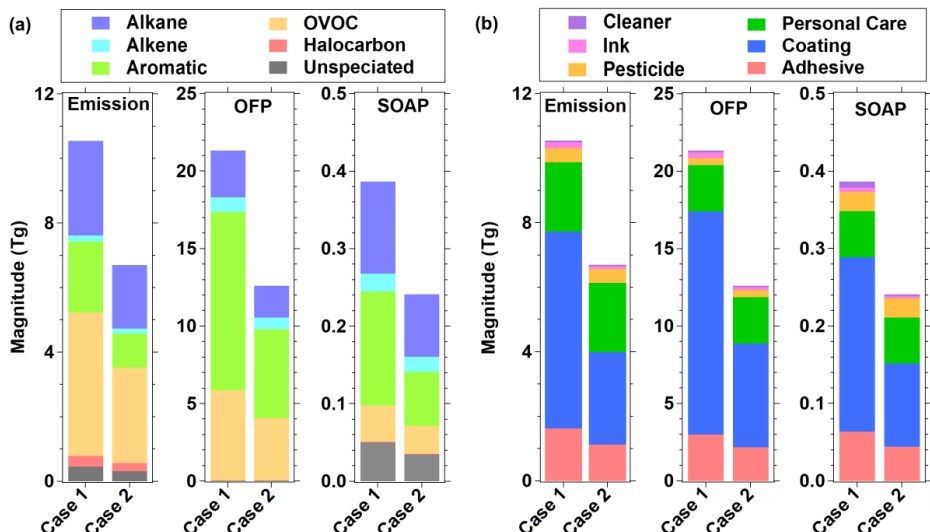

**Figure 9.** Contributions from (a) different source categories and (b) different NMVOCs groups to emissions, OFP, and SOAP. Case 1: emissions in 2017, Case 2: emissions in 2017 after solvent-based products replaced by water-based products.