# Peer review of "A mass balance-based emission inventory of non-methane volatile"

_Atmospheric Chemistry and Physics, 2020_

## Author Comment (AC1)

**Responses to Reviewers' Comments**

In the revised manuscript, we carefully addressed the comments made by the reviewer and clarified the expressions. For ease of review, our responses (in **blue** text) are given point by point to the comments raised by the reviewers (in **black** text). Also, the changes in the manuscript were marked in **red** text.

**Reviewer#1**

This study concerns on the solvent use emissions that has been one of the most popular topics in the VOCs emission research. In particular, the volatile chemical products (VCPs) are considered as the emerging source of urban NMVOCs. However, understanding of NMVOCs emissions from VCPs are still lacking in China. This work addressed this important problem by establishing a detailed emission NMVOCs inventory of solvent use (including six VCPs, i.e., coatings, adhesives, inks, pesticides, cleaners and personal care products) for China based on the mass balance technique. The authors found that NMVOC emissions from solvent use increased rapidly from 2000 to 2014 but leveled off thereafter due to control measures implemented on the solvent-related factories in China. Personal care products become an increasing important source NMVOCs. Speciated emissions, OFP and SOAP as well as the comparisons with previous studies are also analyzed in detail. Overall, this manuscript is well presented and within the scope of ACP. The methods are solid, and the results are informative. It can be accepted for publication after the following comments are addressed.

Reply: We would like to thank the reviewer's valuable and helpful comments. We considered the comments carefully with point-to point responses as follows.

1. Line 122-123: Why the authors considered the six types of organic solvent products. Are there any other products that could be the sources of NMVOCs?

Reply: Thanks for the comment. The six types of the organic solvent products, i.e., coatings, adhesives, inks, pesticides, cleaners and personal care products are most widely used in the industrial and residential activities. They are in line with the recent study in the United States by McDonald et al. (2018). Some other solvent products are also used in pharmaceutical production

and cooking. However, their emissions are minor (<5%) compared with those considered in this study (Wei et al., 2008). Therefore, the six types of solvent product with available estimation parameters and activity data are estimated for China.

Reference

Wei, W., Wang, S. X., Chatani, S., Klimont, Z., Cofala, J., and Hao, J. M.: Emission and speciation of non-methane volatile organic compounds from anthropogenic sources in China, Atmospheric Environment, 42, 4976-4988, https://doi.org/10.1016/j.atmosenv.2008.02.044, 2008.

2. Line 139: In equation (1), emissions of S/IVOC are calculated. Are the NMVOCs included S/IVOCs in this study? Please clarify across the entire manuscript.

Reply: Yes, the S/IVOC emissions are calculated. In this study, the total NMVOCs in products is divided into VOCs and S/IVOCs, considering volatilization of VOCs and S/IVOCs respectively. This is in line with classification in the previous study (McDonald et al., 2018). To clarify the VOCs and S/IVOCs emissions, we added the related contents as follows: "The total NMVOCs emissions can be divided into VOCs and S/IVOCs according to Equation (1), contributing 93% and 7%, respectively (Figure 4a). Among the solvent use products, pesticides emitted the largest contribution (23%) of S/IVOCs, followed by inks (10%), adhesives (10%), coatings (5%), personal care products (5%), and cleaners (4%). This was because of larger S/IVOCs content ($W_{S/IVOC,i}$>20%) in pesticides compared with other products (Table S7). As pesticides emissions were much smaller than coatings and adhesives (Figure 2), total S/IVOCs emissions were not significant (<10% of total NMVOCs emissions). Nevertheless, estimates of S/IVOCs emissions exhibit large uncertainties because of the lack of local measurements of S/IVOCs content in chemical products used in China."

Please see Page 11 Line 328-336 in the revised manuscript.

Reference:
McDonald, Brian C., de Gouw, Joost A., Gilman, Jessica B., Jathar, Shantanu H., Akherati, Ali, Cappa, Christopher D., Jimenez, Jose L., Lee-Taylor, Julia, Hayes, Patrick L., McKeen, Stuart A., Cui, Yu Yan, Kim, Si-Wan, Gentner, Drew R., Isaacman-VanWertz, Gabriel, Goldstein, Allen H., Harley, Robert A., Frost, Gregory J., Roberts, James M., Ryerson, Thomas B., and Trainer, Michael: Volatile chemical products emerging as largest petrochemical source of urban organic emissions, Science, 359, 760, 10.1126/science.aaq0524, 2018.

Line 146-147: Why only the control of NMVOCs emissions from industrial solvent use are considered? Are there any other control measures implemented for the residential sectors in China?

Reply: For the industrial solvent use, exhaust treatment facilities are required to be installed after the *Action Plan for Air Pollution Prevention and Control* was issued by the State Council of China in 2013. As a result, the percentage of solvent use industrial facilities with treatment devices ($C_n$ in Equation 1) increases quickly in the recent years driven by the national policy. However, for the residential sectors such as personal care and daily cleaner, end-of-pipe treatment are hardly implemented in China. Therefore, control of industrial solvent use is considered in this study. We added the following discussion to clarify this problem: "Only end-of-pipe control of NMVOCs from industrial solvent use is considered in this study, while residential emissions such as personal care products and daily cleaners, VOCs treatment is not implemented in residential and commercial buildings in China."

Please see Page 5 Line 149-152 in the revised manuscript.

Line 367-368: "Ink emissions were much higher in MEIC, while similar results were found for Sun EI and this study (Figure 7b)." What are the reasons for these differences?

Reply: The reason is mainly because different emission factors were used. In the Sun EI, inks are divided into conventional inks and new-type inks, with their emission factors of 750g/kg and 100g/kg, respectively. Our work divided inks into different categories, such as solvent-based inks, water-based inks, and other low-emission inks, with their VOC content ($W_{voc}$) of about 60%-63%, 13%-23% and 1%-4% (Table S7). Therefore, high-emission and low-emission inks were considered in Sun EI and our study. However, in the MEIC, a universal value of emission factor (540g/kg) was used for ink emissions. This is the main reason why ink emissions were much higher in MEIC than Sun EI and this study. To further elaborate the differences in ink emissions among the emission inventories, we added the discussion in the revised manuscript: "The reason is mainly because low-emission and high-emission inks were considered in both Sun EI and this study, resulting in much lower estimates than MEIC that adopted a high and universal emission factor." Please see Page 13 Line 407-409 in the revised manuscript.

Line 409-412: "Coatings contribute...solvent-based coatings are dominant ...the wide use of water-based solvent…" The solvent-based coatings and water-based adhesives/inks are large. What are their fractions, respectively? Please clarify here.

Reply: We are sorry for this unclear expression. We revised the sentences as "Coatings contribute most to NMVOCs emission, OFP and SOAP reduction because of the dominant proportion (74%) of solvent-based products in industrial coating. In contrast, the reductions of adhesives and inks emissions, OFP and SOAP are minor due to the wide use of low VOC content products, accounting for 82% of total adhesives and 65% of inks."

Please see Page 14 Line 437-441 in the revised manuscript.

Some other minor comments:

Line 372-374: This sentence is too long.

Reply: The sentence is shortened. It is revised as "Pesticides emissions showed a similar trend between Sun EI and this study, but lower than estimates in MEIC (Figure 7d). There was a significant decrease in 2017 in our work due to that the production of pesticides had decreased and export had increased (Figure S2)." Please see Page 13 Line 413-415 in the revised manuscript.

Line 397: The wording "increase less" is confusing.

Reply: It is removed in the revised manuscript.

Figure 4: The size or positions of the left and right panels can be modified.

Reply: We revised Figure 4 accordingly.

Line 355: "The reasons for …. activity data." This sentence is not completed.

Reply: It is revised "The reasons for the lower estimates in Bo et al. (2008) and Wu et al. (2016) were mainly due to excluding the adhesive emissions and different methods used to estimate personal care emissions (Table S13)." accordingly. Please see Page 13 Line 394-396 in the revised manuscript.

Line 357: between-> from.

Reply: It is revised accordingly.

Line 394-396: "Compared with 2000…industrial process in MEIC in 2017." This sentence is not completed.

Reply: The sentence is removed in the revised manuscript.

---

## Author Comment (AC2)

**Responses to Reviewers' Comments**

In the revised manuscript, we carefully addressed the comments made by the reviewer and clarified the expressions. For ease of review, our responses (in **blue** text) are given point by point to the comments raised by the reviewers (in **black** text). Also, the changes in the manuscript were marked in **red** text.

**Reviewer #2**

This manuscript presents a long-term NMVOCs emission inventory of solvent use during 2000-2017 in China. Based on a mass (material) balance method, NMVOCs emissions were estimated for six categories, including coatings, adhesives, inks, pesticides, 35 cleaners and personal care products. This paper deal with an important issue of NMVOCs emissions from solvent use, which is a major precursor of ozone and fine particle pollution. Strength of this work is to use of direct activities of VOC-containing product consumption statistics for mass balance-based estimation methodology(eq. 1). Several weaknesses, however, are also found in this paper such as, use of national level statistics, lack of I/S VOC result, non-sector specific control application, and weak discussion in general. I, therefore, think this manuscript need to be improved further to be considered for publication. Followings are my review points and suggestions.

Reply: We would like to thank the reviewer's valuable comments and suggestions. We considered the comments carefully and addressed them point by point as follows.

**[Major Comments]**

1.  Page 5, 12, 13 : Estimation of VOCs emissions using production-consumption data (i.e. mass balance-based estimation) should be more accurate way of estimating VOC emissions, compares to EF-based estimation. The inter-comparison to the various EF-based results (e.g. MEIC, Sun et al., etc.) show higher or lower level of agreement depend on the sectors. I strongly recommend authors to discuss deeper for reason of different agreement levels by sector. If possible, subsector level discussion using Table S13 would help understanding differences among EF-based vs. MB-based approach. I think it would be one of the most important knowledge that this work can contribute.

Reply: Thanks for raising this important issue. The inter-comparison and sub-sector level discussion would help understanding differences among EF-based vs. MB-based approach.

Therefore, we added more discussion on the reasons of differences among the EIs by examining the emission factors, activity data, and source classifications in different studies. Table S13 was also used to facilitate the discussion of subsector-level estimations. Please see some examples of deeper discussion marked in *Red*. Please see Page 12-13 for details.

"Our estimates were peaked in 2014, the same with REASv3.1 whose emissions, however, were much higher. The reason is mainly due to higher emission factors used in solvent use (SLV) and paint use (PAIN) estimates in REASv3.2. Some solvent source categories like pharmaceutical production and edible oil production (Wei et al., 2008) were not included because of lacking estimation parameters such as $Wvoc$ for these sources. However, their contributions are not significant (<5%) to the total solvent use emissions (Wei et al., 2008). Emissions in EDGARv4.3.2 were significantly higher than our work in early 2000s. However, with much higher annual growth rate of 12% in our work, emissions surpassed those in EDGARv4.3.2 after 2011. Different activity data were used in EDGARv4.3, which was the main reason for the nearly linear increase of solvent use emissions. Compared with the domestic long-term EIs in China, our results were much higher than Sun EI (from 1.6 Tg in 2000 to 5.0 Tg in 2015; 8%) but very close to MEIC (from 2.3 Tg in 2000 to 11.9 Tg in 2017; 10%). The reason for the lower emissions in Sun EI is because of lower EFs, for example, 80 g/kg in Sun EI compared with 620 g/kg in MEIC for architecture interior wall coating (Table S13). Adhesive emissions were not calculated in Sun EI, which was also an important difference. MEIC showed continuously increasing trend after 2014 but a plateau of NMVOCs emissions was found in this study. It is probably because MEIC did not consider the control of NMVOCs in recent years. For the single year estimates, Bo et al. (2008) and Wu et al. (2016) were lower while Wei et al. (2008) was higher than our results.. The reasons for the lower estimates in Bo et al. (2008) and Wu et al. (2016) were mainly due to not including the adhesive emissions, and different methods used to estimate personal care emissions (Table S13). EFs of solvent-based adhesives and inks in Wei et al. (2008) were higher than estimation parameters in our work. Pharmaceutical production and edible oil production were included in Wei et al. (2008) but not in our work. Different types of activity levels and emission factors also resulted in the discrepancy in EIs. In general, different source categories, EFs, and activity data collectively contribute to the differences among the EIs (Table S13)."

Please see Page 12-13 Line 376-400.

2. Page 6 and 10 : Province level emissions would better be presented as a map with provincial VOCs composition graphs on it, in addition to Figure 3. Considering VOCs' atmospheric lifetime, the level of spatial detail(i.e. national level) in emission estimation is limited. Since the top-down allocation(from national to provincial) using the proxy data(Table S7) is not quite accurate methodology, authors need to discuss more on limitations introduced by national level estimation then downscaling, not directly estimated using local(provincial) statistics as in some EF-based estimation.

Reply: As suggested, we added a map with provincial VOCs emissions and compositions in Figure 3.

[Figure]

**Figure 3.** (a) Spatial distributions of solvent use NMVOCs emissions in China and (b) their source contributions in different provinces in 2017.

More discussion on limitations of the using the proxy data to downscale from national to provincial emissions were also added in the revised manuscript. "There are limitations of using the proxy data to downscale from national to provincial emissions. For example, the sales value of the solvent products cannot fully represent the locations of solvent use processes. Some products might export from the manufacturing province to other provinces. This introduces the uncertainty in the spatial distribution of the solvent use VOCs emissions. Note that local (provincial) statistics for all the solvent use products are still not comprehensively available in China. Nevertheless, direct estimates using local (provincial) statistics could reduce the errors from downscaling."

Please see Page 7 Line 193-199 in the revised manuscript.

3. Page 7, 10, 11, Text S1 : Chemical speciation should be a very important part of the VOCs estimation. The way they were estimated, however, are not clearly presented and discussed in this manuscript. Authors are required to expand Text S1 and Figure S3-S12 to state more detail data and procedure for the chemical speciation, which can support species-based results in the page 11.

Reply: We extended the description of the procedure and data source for the chemical speciation in Text S1. A flow chart of the step-by-step procedure was added and Figure S3-S12 were elaborated. The more detailed discussion is as follows:

"Source profiles of solvents use used in this study are obtained by combining the domestic and foreign profiles. This compilation of the profiles can make use of the local measurements in China as well as include more comprehensive species, such as S/IVOCs being listed in the foreign profiles. The procedure involves four steps as shown in the following flow chart.

**Step 1: Averaging the profiles**

[Figure]

Step 1: A new domestic or foreign profile is formed by averaging weight percentages of NMVOCs groups from multiple source profiles. If some source profiles have OVOC, the treatment of OVOC followed the methods in Wu and Xie (2017) and Li et al. (2014) by averaging the NMHCs and OVOCs proportions, respectively.

Step 2: Common species in the domestic and foreign profiles are identified. Common species may account for different proportions in the domestic and foreign profiles. For example, common species account for b% in the average domestic profile, while B% in the average foreign profile. The remaining unique species account for a% in the domestic profile and C% in the foreign profile. Here a%+b%=100% and B%+C%=100%.

Step 3: We calculate proportion (b%) of common species in the domestic profile, then scale to proportions (B%) of common species in the foreign profile. At the same time, we scale the proportion of unique species in the domestic profile to be A% (=a÷b×B%). Because the foreign profile generally has more comprehensive species and the common species account for a smaller proportion. In order to include more species, we use the proportion of common species in the foreign profile as a reference for scaling.

Step 4: We integrate proportions of common species (B%), unique species in the domestic profile (A%) and unique species in the foreign profile (C%) into a new and complete

source profile. Finally, the proportions of unique species in the domestic profile, common species in both the domestic and foreign profiles, and unique species in the foreign profiles are A/(A+B+C)%, B/(A+B+C)%, and C/(A+B+C)%, respectively.

Figure S3-S12 show the procedure and data source of source profiles for architectural coating, furniture coating, automobile coating, other coating, offset printing ink, letterpress printing ink, gravure printing ink, other printing ink, shoemaking adhesive and herbicide. Here, we take the architectural coating as an example to elaborate the detailed procedure for the chemical speciation following the four-step procedure (Figure S3).

Firstly, the domestic profiles of Yuan et al. (2010) and Wang et al. (2014) are averaged to form a new domestic profile, while the foreign profile of McDonald et al. (2018) is used.

Secondly, the common species in the domestic profile and foreign profiles are identified, accounting for 88.8% (b%) and 25.4% (B%), respectively. The unique species account for 11.2% (a%) in domestic profile while 74.6% (C%) in foreign profile.

Thirdly, the proportion of common species in domestic profile (88.8%) is scaled to proportion in foreign profile (25.4%; B%). The proportion of unique species in domestic profile is scaled to be 3.2% (= 11.2%÷88.8×25.4%; A%).

Finally, we normalized the proportions of common species, and unique species in the domestic and foreign profiles to generate the integrated profile."

Please see Text S1 in the Supporting Information.

4. Page 5, 6: Estimation of S/IVOCs are presented in the Eq 1. and 2., but not stated further in the manuscript. Since the importance of I/SVOCs emissions are growing, I recommend adding contents and discussions for them.

Reply: As suggested by the reviewer, we added the contents and discussions for S/IVOCs. The contributions of VOCs and S/IVOCs were added in Figure 4 in the revised manuscript. "The total NMVOCs emissions can be divided into VOCs and S/IVOCs according to Equation (1), contributing 93% and 7%, respectively (Figure 4a). Among the solvent use products, pesticides emitted the largest contribution (23%) of S/IVOCs, followed by inks (10%), adhesives (10%), coatings (5%), personal care products (5%), and cleaners (4%). This was because of larger S/IVOCs content ($W_{S/IVOC,i}$>20%) in pesticides compared with other products (Table S7). As pesticides emissions were much smaller than coatings and adhesives (Figure 2), total S/IVOCs emissions were not significant (<10% of total NMVOCs

emissions). Nevertheless, estimates of S/IVOCs emissions exhibit large uncertainties because of the lack of local measurements of S/IVOCs content in chemical products used in China."

Please see Page 11 Line 328-336 in the revised manuscript.

[Figure]

**Figure 4.** (a) Contributions of VOCs and S/IVOCs, (b) NMVOCs functional group pattern, and (b) the top 10 species in NMVOCs emissions in 2017 from solvent use.

5. Page 8, 9, 13, Figure 2, 9 : VOC emission control stated only for industrial production (i.e. industrial process) in "3.1 Control of NMVOCs emissions" section which is limited. Analysis of emissions control in manufacturing industry would better be sorted by not only control technology (Table S8) but type of industry (i.e sub-sectors). Since much of emissions coming from the consumption process, authors need to add contents and discussion on this process. For example, application of water-based vs. solvent-based paint with respect to national/regional control polices. I suggest to merge control-related contents in the section 4.3 (also, in Figure 9), need to be merged in the section "3.1 Control of NMVOCs emissions."

Reply: We thank the reviewer for pointing out that the emission control would be also sorted by industrial sectors/sub-sectors. However, the information of the control technology for specific sectors are rather limited in China. We have put efforts in looking for more data about the control technology by sub-sector, However, no official and published data are available. We therefore cannot make a more detailed classification at this moment. Nevertheless, more field survey and data collection are needed in our next-step research. Regarding the application of water-based and solvent-based paint with respect to national/regional control policies, as we know, no specific differences are found in applying the control measures in China. This is mainly because control technologies of VOCs emission on solvent use sectors are still developing and not mature in China. As suggested, we added the discussion in the revised manuscript with "control measures are required to be installed for NMVOCs emitting industrial facilities related to solvent use in China. The percentage of solvent use industrial facilities with treatment devices ($C_n$ in Equation 1) increases quickly in the recent years. Note that the NMVOCs control technology is still developing and not mature in China. At this time, limited information is available to determine control technology by specific sectors and solvent products." Please see Page 8 Line 240-245 in the revised manuscript.

For the section 3.1 Control of NMVOCs emission, we mainly focus on control measures/technologies on the NMVOCs in recent years due to implementations of the national policy- *Action Plan for Air Pollution Prevention and Control* issued by the State Council of China in 2013. This could also help explain why the solvent use NMVOCs emissions leveled off in recent years (Figure 2).   In contrast, in the section 4.3 Implications for NMVOCs control, we

are concerning the reduction potential by replacing the solvent-based product with water-based product. Therefore, we make a comparison between two scenarios - Case 1: emissions in 2017, Case 2: emissions in 2017 after solvent-based products replaced by water-based products. This result could give implications for China's policy makers to consider the benefits of using water-based products and in turns the effectiveness in OFP and SOAP reduction. We therefore would like to remain these two parts in the revised manuscript.

6.  Page 13: Contents in the "4.2. Comparison with other source" mostly discuss about the cross-sectoral importance changes using other references. I would suggest to shrink and move to introduction and/or conclusion chapters.

Reply: We shrank the discussion of *4.2 Comparison with other source* and moved the content to the *4 Conclusion.* Nevertheless, we kept Figure 8 and moved it to Figure S14 in the Supporting Information. Please see the revised discussion with "Emissions from solvent use grew quickly (with an over five-fold increase) during 2005-2013 and reached a plateau after 2014, which we attribute to the significant industrial expansion in China over the past decades, and effective control on solvent use in recent years (Figure S13). In contrast, combustion and transportation exhibited a decline in the past decade, mainly because of the stringent control of NMVOCs from fuel combustion by industry and on-road vehicles." in Page 15 Line 466-471 in the revised manuscript.

**[Minor Comments]**

Page 6 : Authors state that "Uncertainty is set to be ±30% if data are directly from official statistics; uncertainty is assumed to be ± 80% if activity data is estimated from other statistical information or reports." How could authors set these values? Form Wei et al, 2011a? Please elaborate more.

Reply: Yes, the values used in this study followed the suggestions by Wei et al. (2011a). They established an evaluation system for uncertainty in activity data, see the table below. We added more elaboration with "We estimate the uncertainty by combining the coefficients of variation (CV, or the standard deviation divided by the mean) of the activity data and the VOCs and S/IVOC contents (Street et al., 2003). According to the accuracy and reliability of the activity data, five-tier evaluation system for uncertainty in activity data was established by Wei et al.

(2011a) as shown in Table S6. We set the uncertainty as ± 30% if data are directly from official statistics and ± 80% if activity data is estimated from other statistical information or reports." in the revised manuscript. Please see Page 6 Line 180-186 in the revised manuscript.

Table S6 Five-tier evaluation system for uncertainty in activity data (Wei et al., 2011a)

| Tier | Data source | Uncertainty |
|------|-------------|-------------|
| I | Directly from official statistics | ± 30% |
| II | Estimated from other statistical information or reports; The data is strongly related; The statistical information or reports is reliable. | ± 80% |
| III | Estimated from other statistical information or reports; The data is strongly related; The statistical information or reports are less reliable. | ± 100% |
| IV | Estimated from other statistical information or reports; The data is less related; The statistical information or reports is reliable. | ± 150% |
| V | The data is less related; The statistical information or reports is less reliable. | ± 300% |

Reference:

Wei, W., Wang, S., and Hao, J.: Uncertainty analysis of emission inventory for volatile organic compounds from anthropogenic sources in China (in Chinese). Environmental Science, 32, 305-312, 2011a.
Streets, D. G., Bond, T. C., Carmichael, G. R., Fernandes, S. D., Fu, Q., He, D., Klimont, Z., Nelson, S.M., Tsai, N.Y., Wang, M.Q., Woo, J.H., and Yarber, K. F.: An inventory of gaseous and primary aerosol emissions in Asia in the year 2000. Journal of Geophysical Research: Atmospheres,108(D21), 8809, doi:10.1029/2002JD003093, 2003.

Page 26: Description for (a) and (b) need to be switched.

Reply: Thanks for the suggestion. We switched the description for (a) and (b).

---

## Author Comment (AC3)

**Responses to Reviewers' Comments**

In the revised manuscript, we carefully addressed the comments made by the reviewer and clarified the expressions. For ease of review, our responses (in **blue** text) are given point by point to the comments raised by the reviewers (in **black** text). Also, the changes in the manuscript were marked in **red** text.

**Reviewer #3**

This study is very interesting because it uses a unique approach to estimate NMVOC emissions in China. In particular, the approach of estimating VOC emissions from adhesives is excellent in terms of the points of view.

Reply: We appreciated it very much for the reviewer's valuable comments. We addressed the concerned raised below to improve the quality of our manuscript. Please see following responses.

It would be good to work on deriving the VOC composition from the literature values, as shown carefully in Figures S3 -12.

Reply: Thanks for the comment. Due to lack of comprehensive VOC source profiles for different categories of the solvent use in China, we therefore derived the VOC compositions by combining available literature values to reduce the uncertainty and avoid bias from individual-specific measurement. See response to Reviewer #2, Comment #3 for more details on our VOC speciation procedure.

It is still necessary to examine the details in order to use these results as input data for an atmospheric model, like CMAQ, but I think it will be useful enough for discussions on understanding the NMVOC emissions of solvents from China.

Reply: We totally agree with the reviewer's that the detailed VOC compositions are urgently needed in the air quality modeling, like using CMAQ as mentioned. However, VOC speciation remains largely uncertain because of lacking local China source profiles. Our study is trying advance our understandings on this aspect. Details on the NMVOC compositions of solvent use are discussed in *Section 3.4 Speciated NMVOCs emissions, OFP and SOAP*. This could provide

implications for atmospheric modeling results. Nonetheless, model-ready emission inventories are still needed in our future studies so as to facilitate air quality modeling.

However, with regard to IVOC, it is a pity that only parameters related to IVOC are listed in Table S6.

Reply: Thanks for the comment. We added the results and discussion related to S/IVOCs in the revised manuscript as follows: "The total NMVOCs emissions can be divided into VOCs and S/IVOCs according to Equation (1), contributing 93% and 7%, respectively (Figure 4a). Among the solvent use products, pesticides emitted the largest contribution (23%) of S/IVOCs, followed by inks (10%), adhesives (10%), coatings (5%), personal care products (5%), and cleaners (4%). This was because of larger S/IVOCs content ($W_{S/IVOC,i}$>20%) in pesticides compared with other products (Table S7). As pesticides emissions were much smaller than coatings and adhesives (Figure 2), total S/IVOCs emissions were not significant (<10% of total NMVOCs emissions). Nevertheless, estimates of S/IVOCs emissions exhibit large uncertainties because of the lack of local measurements of S/IVOCs content in chemical products used in China." Please see Page 11 Line 328-336 in the revised manuscript.

In addition, authors mentioned VOC source categories that are not included in this study (L359-360). Do the authors not include them in the total amount as reference or reference data? I think that is one of the reasons of the differences between the VOC emissions from the solvent of REAS v3.1 and this study.

Reply: Yes, the source categories such as pharmaceutical production and edible oil production were not included in this study because of lacking estimation parameters such as *Wvoc* for these sources. It could be one of the reasons for the lower estimates of VOCs emissions in this study than other studies. However, these categories are not significant, contributing less than 5% of the total solvent VOCs emissions (Wei et al., 2008). The difference between REAS v3.2 and this study is probably due to the different emission factors used in the estimation. We added the related discussion "The reason is mainly due to higher emission factors used in solvent use (SLV) and paint use (PAIN) estimates in REASv3.2. Some solvent source categories like pharmaceutical production and edible oil production (Wei et al., 2008) were not included because of lacking estimation parameters such as *Wvoc* for these sources. However, their contributions

are not significant (<5%) to the total solvent use emissions (Wei et al., 2008). in the revised manuscript. Please see Page 12 Line 377-382.

Reference:
Wei, W., Wang, S. X., Chatani, S., Klimont, Z., Cofala, J., and Hao, J. M.: Emission and speciation of non-methane volatile organic compounds from anthropogenic sources in China, Atmospheric Environment, 42, 4976-4988, https://doi.org/10.1016/j.atmosenv.2008.02.044, 2008.

There are some other things that I noticed:

**"3.1 Control of NMVOCs emissions" :**

Please indicate which of the six categories you are considering applying industrial solvent emission control to. Also, please indicate whether the values are uniform or individually set for all applied categories. For coating, what do the authors think of architectural coating for emission control?

Reply: We considered the emission control for sub-categories of industrial solvent use processes, including industrial solvent use included coatings except architectural coatings, inks, industrial adhesives (woodworking, paper converting, shoemaking, fiber processing, packaging, and labelling), and industrial detergents(Figure S7). For the architectural coating, we did not consider the emission control in this study because VOCs treatment devices are hardly used in China. We added the description in the revised manuscript to clarify. "…control of NMVOCs emissions from the solvent use industrial sources, including coatings except architectural coatings, inks, industrial adhesives (woodworking, paper converting, shoemaking, fiber processing, packaging, and labelling), and industrial detergents considered in this study(Table S7)." Please see Page 8 Line 238-245 in the revised manuscript.

**L53:** Please write the full terms of OFP and SOAP in the abstract.

Reply: The full terms of OFP and SOAP are added in the revised manuscript.

**L122-125:** I think it is better to write somewhere that "Level 1" refers to the six categories (coatings, inks, adhesives, pesticides, cleaners and personal care).

Reply: We added "Level 1" in the revised manuscript.

**L125-128:** I think it is better to explain the subcategories in the order they appear in L122-123.

Reply: We explained the subcategories in the same order as they appeared for the first time.

Please see Page 5 Line 125-131 in the revised manuscript.

**L161-164:** I think the Wvoc listed in Table S6 was calculated by the authors based on Table S1-S5, but they need to mention that clearly.

Reply: Thanks for pointing out the problem. We added the explanation in the revised manuscript as follows: "Table S1-S5 listed the $W_{voc}$ for different sub-categories of coatings, inks, adhesives, pesticides and cleaners, and personal care products, respectively. Taking architectural coatings as an example, the VOCs content of solvent- and water-based coatings are obtained on two national standards (GB) for VOC emission restrictions in China-GB18582-2008 and GB24408-2009. Averages were used when several values are available from different regions of China and data sources. Those categories lacking $W_{voc}$ were approximated by the values from similar sources." Please see Page 6 Line 167-173 in the revised manuscript.

Only insecticide is shown for pesticides in Table S4. Table S6 lists Insecticide, Herbicide, Bactericide and Other. How did the authors set the VOC ratio for pesticide components other than insecticide?

Reply: We are sorry that only the $W_{voc}$ of insecticide is available from the literature. Accordingly, this value was taken to appropriate the VOC contents of Insecticide, Herbicide, Bactericide and Other in Table S7. We added the explanation with "Those categories lacking $W_{voc}$ were approximated by the values from similar sources" in the revised manuscript.

**L163:** It would be nice to note that GB18582-2008 and GB24408-2009 are the national standards (GB) for volatile organic compound (VOC) emission restrictions in China.

Reply: We added the explanation in the revised manuscript.

**L170:** Eqation => Equation

Reply: It is corrected.

**L193-194:** The authors applied foreign profiles to the products but are the source of the foreign profiles "Li et al., 2014".? Do the authors see "Lie et al., 2014" as a methodology reference? Please write the reference of the foreign profiles.

Reply: Sorry for the confusing expression. We mainly used the foreign profiles from the US (McDonald et al., 2018), and we cited Li et al., 2014 as a methodology reference for combining the local and foreign profiles. We therefore revised the expression as "Source profiles of solvents use used in this study are obtained by combining domestic profiles (e.g., Wang et al., 2014b; Yuan et al., 2010) and foreign profiles (McDonald et al., 2018), following the methods proposed by Li et al., (2014)." The references of domestic source profiles as well as the foreign profiles are illustrated in the Figure S3-S12. Please see Page 7 Line 205-207 in the revised manuscript.

**L305:** Please write the full terms of OVOC.

Reply: We added the full term of OVOC with "oxygenated VOC" in the revised manuscript.

**L323:** SOA => SOAP

Reply: It is corrected.

**L340:** The reference for REAS v3.1 is incorrect. This reference is REASv2.

Reply: Sorry for the inappropriate citation. We revised the reference with "Kurokawa, J. and Ohara, T.: Long-term historical trends in air pollutant emissions in Asia: Regional Emission inventory in ASia (REAS) version 3, Atmos. Chem. Phys., 20, 12761–12793, https://doi.org/10.5194/acp-20-12761-2020, 2020."

**Figure 6:** This REAS v3.1 NMVOC is the sum of PAINT and SLV, but the decrease from 2014 to 2015 does not seem to be so large. Since the numerical value of REAS v3.2 has been released, I think it is good to replace it.   URL: https://www.nies.go.jp/REAS/

Reply: We replace the REAS v3.1 with REAS v3.2 accordingly.

**Figure 5:** How did the authors decide on the SOAP for unspecified VOCs?

Reply:  We followed the method in McDonald et al. (2018) to allocate a value of 0.11 (n-tetradecane) for the unspeciated VOCs. Please also see parts of Table S8 in McDonald et al.

(2018) as follows. We added a list of VOCs and S/IVOCs species and their MIR and SOA yield in Table S9.

| No. C | No. O | CARB Compound [a] (surrogate) | TVOC % | R$_{OH}$ % | SOA % | k$_{OH}$ [b] ppb$^{-1}$ s$^{-1}$ | SOA Yield g g$^{-1}$ | log C* [c] µg m$^{-3}$ | Indoor[d] fraction |
|---|---|---|---|---|---|---|---|---|---|
| 14 | 0 | n-tetradecane | 0.16 | 0.10 | 0.48 | 0.41 | 0.11 ± 0.02 | 5.8 | 0.47 |
| | | *Unspeciated* | | | | | | | |
| -- | -- | diesel | 1.4 | 1.3 | 9.1 | 0.28 | 0.23 ± 0.04 | 6.5 | 0.39 |
| -- | -- | non-oxy IVOCs (n-tetradecane) | 1.9 | 0.7 | 4.3 | 0.41 | 0.11 ± 0.02 | 5.8 | 0.22 |
| -- | -- | oxy IVOCs (n-tetradecane) | 3.8 | 1.4 | 8.5 | 0.41 | 0.11 ± 0.02 | 5.8 | 0.22 |
| | | ∑ = | 7.2 | 3.4 | 21.9 | | | | |

Reference:

McDonald, Brian C., de Gouw, Joost A., Gilman, Jessica B., Jathar, Shantanu H., Akherati, Ali, Cappa, Christopher D., Jimenez, Jose L., Lee-Taylor, Julia, Hayes, Patrick L., McKeen, Stuart A., Cui, Yu Yan, Kim, Si-Wan, Gentner, Drew R., Isaacman-VanWertz, Gabriel, Goldstein, Allen H., Harley, Robert A., Frost, Gregory J., Roberts, James M., Ryerson, Thomas B., and Trainer, Michael: Volatile chemical products emerging as largest petrochemical source of urban organic emissions, Science, 359, 760, 10.1126/science.aaq0524, 2018.

---

## Referee Report (RR1)

I would like to appreciate the authors for their polite and thoughtful responses to all my points and questions. With these answers, my questions were cleared up. I would also like to thank the other reviewers for their suggestions and the authors for their appropriate replies, which helped me to understand the paper better.

However, there are two points that I would like the authors to correct.
In the description of other VOC estimation results in LINE 372-380, REAS is mentioned, but v3.1 and v3.2 are mixed up. The reference you corrected is for v3.2, and based on the content of the text, it seems that it does not need to be v3.1. Please unify everything with v3.2.
Similarly, for EDGAR, please use v4.3.2 instead of EDGARv4.3 for the added text in L384 to avoid confusion.